# Document Liveness Challenge Dataset (DLC-2021)

**DOI:** 10.3390/jimaging8070181

**Published:** 2022-06-28

**Authors:** Dmitry V. Polevoy, Irina V. Sigareva, Daria M. Ershova, Vladimir V. Arlazarov, Dmitry P. Nikolaev, Zuheng Ming, Muhammad Muzzamil Luqman, Jean-Christophe Burie

**Affiliations:** 1Smart Engines Service LLC, 117312 Moscow, Russia; iv.sigareva@smartengines.com (I.V.S.); d.ershova@smartengines.com (D.M.E.); vva@smartengines.com (V.V.A.); dimonstr@iitp.ru (D.P.N.); 2Federal Research Center “Computer Science and Control” RAS, 119333 Moscow, Russia; 3National University of Science and Technology MISIS, 119049 Moscow, Russia; 4Moscow Institute of Physics and Technology, 141701 Dolgoprodny, Russia; 5Faculty of Mechanics and Mathematics, Lomonosov Moscow State University, 119991 Moscow, Russia; 6Institute for Information Transmission Problems (Kharkevich Institute) RAS, 127051 Moscow, Russia; 7L3i Laboratory, La Rochelle University, 17042 La Rochelle, France; zuheng.ming@univ-lr.fr (Z.M.); muhammad_muzzamil.luqman@univ-lr.fr (M.M.L.); jcburie@univ-lr.fr (J.-C.B.)

**Keywords:** document analysis, document recognition, identity documents, mobile recognition, liveness detection, document anti-fraud, document forgery detection, screen recapture detection, open data

## Abstract

Various government and commercial services, including, but not limited to, e-government, fintech, banking, and sharing economy services, widely use smartphones to simplify service access and user authorization. Many organizations involved in these areas use identity document analysis systems in order to improve user personal-data-input processes. The tasks of such systems are not only ID document data recognition and extraction but also fraud prevention by detecting document forgery or by checking whether the document is genuine. Modern systems of this kind are often expected to operate in unconstrained environments. A significant amount of research has been published on the topic of mobile ID document analysis, but the main difficulty for such research is the lack of public datasets due to the fact that the subject is protected by security requirements. In this paper, we present the DLC-2021 dataset, which consists of 1424 video clips captured in a wide range of real-world conditions, focused on tasks relating to ID document forensics. The novelty of the dataset is that it contains shots from video with color laminated mock ID documents, color unlaminated copies, grayscale unlaminated copies, and screen recaptures of the documents. The proposed dataset complies with the GDPR because it contains images of synthetic IDs with generated owner photos and artificial personal information. For the presented dataset, benchmark baselines are provided for tasks such as screen recapture detection and glare detection. The data presented are openly available in Zenodo.

## 1. Introduction

The growing popularity of mobile services increases the risk of financial and other losses from fraudulent user actions. To reduce the number of illegal actions and comply with the law when using mobile services, it is often required for users to present their identity documents. In the case of remote access via a mobile device, this means receiving and analyzing identity (ID) document images. ID document recognition systems [1,2] are widely used to obtain and check users’ personal information in many applications. At the same time, despite a large number of publications on the topic of ID document recognition, due to legal and ethical restrictions, researchers are constrained by [3] a lack of open datasets that can be used to reproduce and compare results. The absence of open datasets for ID document fraud prevention research inspired us to create a new dataset, called DLC-2021 [4,5,6]. It can be used to establish an evaluation methodology and set up baselines for document image recapture detection, document photocopy detection, and document lamination detection methods.

## 2. Overview

The GDPR [7] and other local laws prohibit the creation of datasets with real ID images. Thus, researchers began to use artificially generated ID document images for open dataset creation [8,9,10,11,12]. As far as we know, printed mock documents are used only in MIDV family datasets, and MIDV-500 was the first [8]. This dataset contained 500 video clips of 50 identity documents, with 10 clips per document type. The identity documents were of different types, and were mostly “sample” or “specimen” documents that could be found in WikiMedia and were distributed under public copyright licenses. The conditions represented in MIDV-500 thus had some diversity regarding the background and the positioning of the document in relation to the mobile capturing process; however, they did not include variation in lighting conditions, or significant projective distortions. MIDV-2019 [9] was later published as an extension of MIDV-500. It contained video clips captured with very low lighting conditions and with higher projective distortions. The dataset was also supplemented with photos and scanned images of the same document types to represent the typical input for server-side identity document analysis systems. MIDV-2020 [10] was published recently to provide variability in the text fields, faces, and signatures, while retaining the realism of the dataset. The MIDV-2020 dataset consists of 1000 different physical documents (100 documents per type), all with unique, artificially generated faces, signatures, and text field data. Each physical document was photographed and scanned, and for each a video clip was captured using a smartphone. The ground truth includes ideal text field values, and the geometrical position of documents and faces in each photo, scan, and video clip frame (with 10 frames-per-second annotation). MIDV-LAIT [11] contains video for ID documents with textual fields in Perso-Arabic, Thai, and Indian scripts.

When using mobile-based ID document recognition systems, the most technically simple and accessible attack methods are different types of rebroadcast attacks [13]. For the DLC-2021 dataset we shot mock documents from the MIDV-2020 collection as originals (Figure 1a) and modeled those types of attacks that remain realistic when using mock documents: capturing a color printed copy of a document without lamination (Figure 1b), capturing a gray printed unlaminated copy of a document (Figure 1c) and capturing a displayed image of a document (Figure 1d).

Thus, all images in the MIDV family of datasets [8,9,10,11] can be considered as images of genuine documents and can be used as negative samples for document fraud detectors.

There are many studies on engineering- [14,15,16,17,18,19,20] and neural-network-based methods [13,21,22,23,24] for screen recapture detection, but most of them are focused on the analysis of natural images publicly available in the following datasets: NTU-ROSE [15], ICL-COMMSP [16,18], and BJTU-IIS [17]. These datasets are captured as photos by high-resolution DSLR cameras.

Document-specific methods for detecting document recapture are based on the latest advances in deep learning. The algorithm proposed in [25] takes advantage of both metric learning and image forensic techniques. The authors considered practical domain generalization problems, such as the variations in printing/imaging devices, substrates, recapturing channels, and document types with a private dataset. The texture and reflectance characteristics of the bronzing region are used as discriminative features to detect a recaptured certificate document in [26]. The dataset used in the study is available upon request.

Thus, for research in the field of document recapture prevention, new specialized open datasets captured with smartphones are required.

## 3. DLC-2021 Dataset Description

The set of 10 ID document types for DLC-2021 (Table 1) coincides with the set of document types in the MIDV-2020 dataset.

For each type of document, eight examples of physical documents were taken. For selected physical documents, color and gray paper hard copies were made by printing without lamination. All color copies and some of the gray copies were cut to fit the original document page shape.

While preparing the DLC-2021 dataset, we focused on video capture. On the one hand, the video stream allows for analysis changes in time, and this provides much more information for assessing liveliness. On the other hand, video frames usually contain compression artifacts that can significantly affect the performance of analysis algorithms.

In general, DLC-2021 follows the structure of the MIDV-2020 folder and files, except for clip names. In DLC-2021, the two-digit document template number clip name is extended with a two-letter video type code (Table 2) and four-digit serial number.

An Apple iPhone XR and Samsung S10 were used for video capturing, as in MIDV-2020. Video clips were shot with a wide-angle camera (Table 3) using a standard smartphone camera application.

To make videos more varied, we used two different frame resolutions (1080 × 1920, 2160 × 3840) and two different frame rates (30, 60 fps) for shooting video clips. Table 4 summarizes the number of video clips by type.

Each clip was shot vertically and was at least five seconds long. Frames were extracted at 10 frames per second using ffmpeg version n4.4 with default parameters, and for the first 50 extracted frames the document position was manually annotated. The annotation file for each clip followed the MIDV-2020 JSON format [10] and was readable with VGG Image Annotator (v2.0.11) [29].

### 3.1. Paper Document Shooting

We captured video with the “original” documents and printed copies under different lighting conditions, such as natural daylight, bright light with deep shadows, artificial light, colored light, low light, and flashlight. The color characteristics of document images varied significantly under different lighting and capture conditions (Figure 2).

Low or uneven lighting and white balance correction algorithms inappropriate for the lighting conditions dramatically affect color reproduction and complicate the process of distinguishing color documents from gray copies (Figure 3) without specialized color correction algorithms, such as that in [30].

To achieve greater realism of the video, various document occlusions were made on some of the clips (Figure 4), such as holding the document with fingers, and a brightly colored object in the document area.

In the task of detecting gray copies, such partial occlusion can create additional difficulties, as it can lead to an increase in the color diversity of pixels in the document area.

Since ID documents are used regularly, manufacturers protect them from dirt, creases, and other damage by using a special protective coating or lamination. Such a coating can preserve the integrity of documents for a long time from various environmental influences and also significantly complicate attempts to change the content of the document, for example, such as replacing a photo. However, laminated documents can easily introduce reflection and saturation phenomenon, especially when a strong illuminant such as a flash, a fluorescent lamp, or even the sun lights the document during the video-capturing process. Figure 5 shows some images extracted from a video captured with a smartphone.

Strong reflections on the smooth surface of the laminated documents can partially or totally hide the content of the document, making it impossible to analyze the pictures or to extract the text. In addition, the shape and the size of the area of reflection may vary depending on the orientation of the document relative to the smartphone lens. On the one hand, these variations are challenging for detection, segmentation, and recognition algorithms. On the other hand, the analysis of the shape and consistency of changes in highlights and scene geometry can serve as an important indicator of the liveliness of a document. For example, exploiting the camera flashlight during the capture process creates semi-controlled lighting conditions in which laminated and unlaminated documents in some cases can be differentiated more robustly (Figure 6).

### 3.2. Screens Shooting

For screen recapture, we used two office desktops and two notebook LCD monitors. Figure 7 shows samples from the template image and video for original and screen-recaptured cases.

It should be noted that the documents themselves may have a complex textured page background, for example, when using document-protection technologies such as guilloche. Another interesting case is textured scene objects, or even the LCD screen behind the document. In such cases, moiré and other recapture artifacts can also occur outside the document zone when the original document is captured with a digital camera.

## 4. Experimental Baselines

While the main goal of the paper is to present a document liveness challenge dataset, DLC-2021, in order to provide a baseline for future research involving the dataset, in the following sections several benchmarks using DLC-2021 will be presented. As a baseline method we chose Convolutional Neural Networks (CNNs) in view of the fact that CNNs show state-of-the-art results in image classification tasks. In our experiments we used the Keras (2.6.0) library from the Tensorflow (2.8.0) [31] framework and Scikit-learn (1.2.0) library [32]. Scripts, instructions, and pre-trained models to reproduce our experiments can be downloaded from [4].

### 4.1. Screen Recapture Detection

For screen recapture detection, we used a classification CNN model based on ResNet-50 architecture [33] pre-trained on ImageNet weights from TensorFlow Model Garden. We froze the first 49 layers and reduced the number of the last softmax layer outputs to 2. For learning, we used the binary cross-entropy loss function and Adam [34] optimizer with a constant learning rate (lr=0.1).

The screen recapture detector classifies 224×224 patches cut from the center of the document on the original frame. Negative samples are collected from MIDV-2020 images. To collect positive samples, we cropped and manually labeled patches from DLC-2021 recaptured images of Spanish IDs, Latvian passports, and internal passports from Russia. The training set consisted of 19,543 positive and 25,980 negative samples.

The validation dataset contained 11,009 positive and 16,264 negative samples formed from original document images and recaptured images for other DLC-2021 document types. Table 5 shows results from the validation dataset for CNN-based and Scikit Dummy Classifier detectors with different strategies: “constant” (generates constant prediction), “stratified” (generates predictions with respect to the balance of training set classes), and “uniform” (generates predictions uniformly at random). Results for “stratified” and “uniform” strategies were averaged over 10 runs with different seed values, and the standard deviation values are shown in the table.

Most of the false-positive (FP) errors were caused by documents having complex textured backgrounds and compression artifacts, as shown in Figure 8.

### 4.2. Unlaminated Color Copy Detection

The presence of glare is the most evident feature of laminated documents. An unlaminated color copy detector classifies projective undistorted images by frame markup and scaled-down document images. The ResNet-50-based CNN detector showed a steady trend of overfitting, so a more simple architecture, as presented in Table 6, was used.

The CNN-based detector was trained on gray images scaled down to 76×76 with a binary cross-entropy loss function and Adam optimizer (learning rate =0.05). Early stopping and data augmentation (brightness distortion with range [0.9,1.1]) were used to avoid overfitting.

The training dataset was collected from manually labeled MIDV-500 and MIDV-2020 images and contained 29,564 positive and 7544 negative samples. The validation dataset was collected from manually labeled DLC-2021 clip images (or and cc types) and contained 34,607 positive and 3388 negative samples. Table 7 shows the results from the validation dataset for CNN-based and Scikit Dummy Classifier detectors.

### 4.3. Gray Copy Detection

Projective undistorted document images were used for classification. Positive samples in the training set were collected from gray copy clips of Azerbaijani passports, Finnish ID cards, and Serbian passports. Negative samples in the training set were obtained from MIDV-2020. The training set contained 3492 positive and 1000 negative samples. The validation set contained copied grey clips for all other types of documents and original document clips from DLC-2021 (10473 positive and 16264 negative samples).

All experiments with ResNet-50-like models (similar to Section 4.1) and more simple CNN models (similar to Section 4.2) failed. Models either did not train at all or were overfitted. One reason for this result is that CNNs are sensitive to intensity gradient features but ignore color features. Since the development of a more sophisticated CNN architecture is beyond the scope of this article, as a simple baseline, we examined the Scikit Dummy Classifier detector on the validation dataset, as shown in Table 8.

## 5. Conclusions

In this paper, we presented the DLC-2021 dataset containing video clips of mock “real” identity documents from the MIDV-2020 collection and three types of popular rebroadcast attacks: capturing a color printed copy of a document without lamination, capturing a gray printed unlaminated copy of a document and capturing a displayed image of a document. Video was captured using modern smartphones with different video quality, and a wide range of different real-world capturing conditions were simulated. Selected video frames were accompanied by the geometric markup of the outer borders of the document.

Using mock documents from the MIDV-2020 collection as targets for shooting DLC-2021 video makes it easy to use field values and document geometry markup from MIDV-2020 templates. The prepared open dataset can be used for other ID-recognition tasks:–Document detection and localization in the image [35,36,37];–Document type identification [35,37];–Document layout analysis;–Detection of faces in document images [38] and the choice of the best photo of the document owner [39];–Integration of the recognition results [40];–Video frame quality assessment [41] and the choice of the best frame [42].

As the videos were captured with two different smartphones, the DLC-2021 dataset can be used for sensor noise (PRNU)-based method analysis.

In the future, we plan to expand the DLC dataset with more screen types and devices for shooting, as well as increase the variety of document types.

Regarding ethical AI, the published dataset has no potential to affect the privacy of individuals regarding personal data, since all documents are synthetic mock-ups and comply with the GDPR.

The authors believe that the provided dataset will serve as a valuable resource for ID document recognition and ID document fraud prevention, and lead to more high-quality scientific publications in the field of ID document analysis, as well as in the general field of computer vision.

## Figures and Tables

**Figure 1 jimaging-08-00181-f001:**
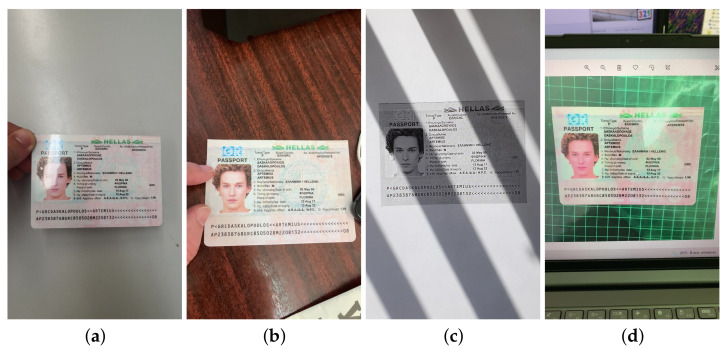
Types of video in DLC-2021 dataset: (**a**) original document, (**b**) unlaminated color copy, (**c**) unlaminated gray copy, and (**d**) document recaptured from screen.

**Figure 2 jimaging-08-00181-f002:**
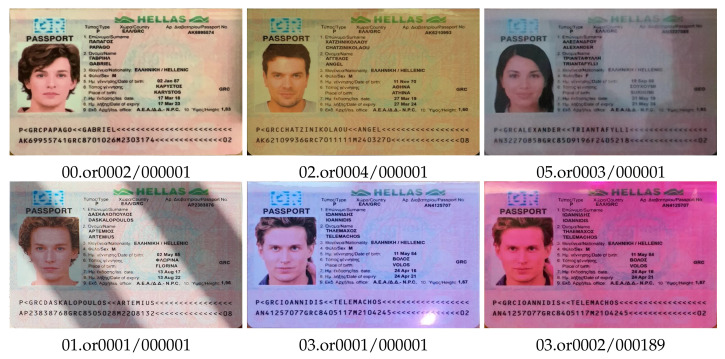
Color variation of original document images for a variety of capture conditions, type grc_passport.

**Figure 3 jimaging-08-00181-f003:**
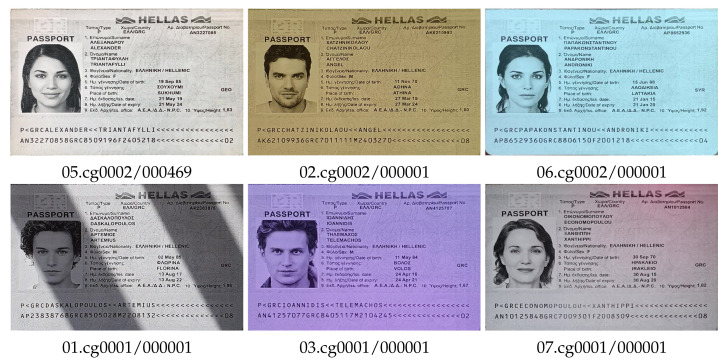
Color variation in gray copies of document images for a variety of capture conditions, type grc_passport.

**Figure 4 jimaging-08-00181-f004:**
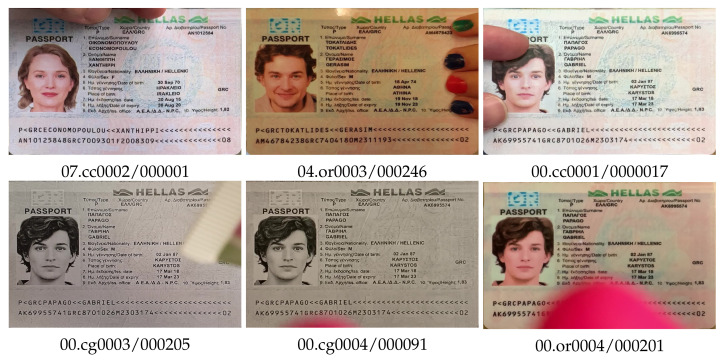
Overlapping variation of document images, type grc_passport.

**Figure 5 jimaging-08-00181-f005:**
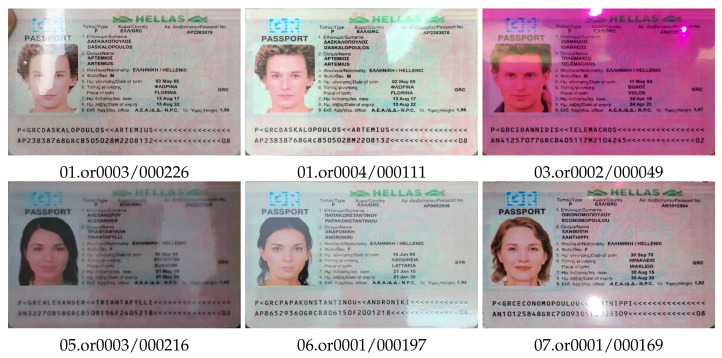
Reflections caused by the lighting condition on the surface of laminated documents, type grc_passport.

**Figure 6 jimaging-08-00181-f006:**
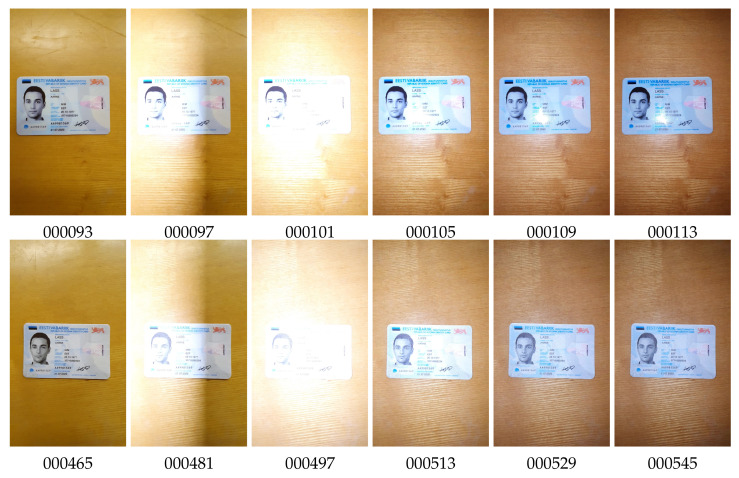
Frames with reflections caused by a flashlight for a laminated document, clip est_id/04.or0004 (**top**), and unlaminated document, clip est_id/04.cc0009 (**bottom**).

**Figure 7 jimaging-08-00181-f007:**
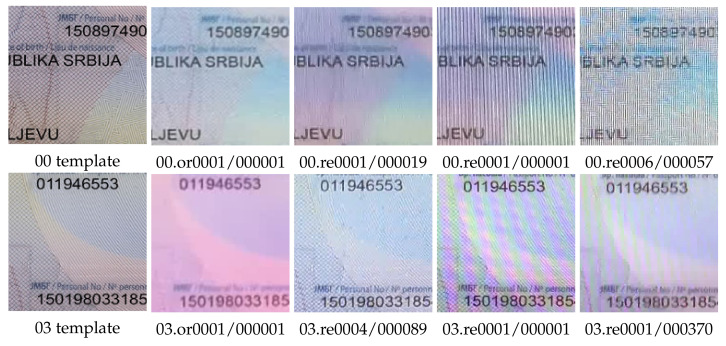
Zones from geometry-normalized images for template (column 1), original (column 2) and recaptured documents (column 3–5), type srb_passport.

**Figure 8 jimaging-08-00181-f008:**
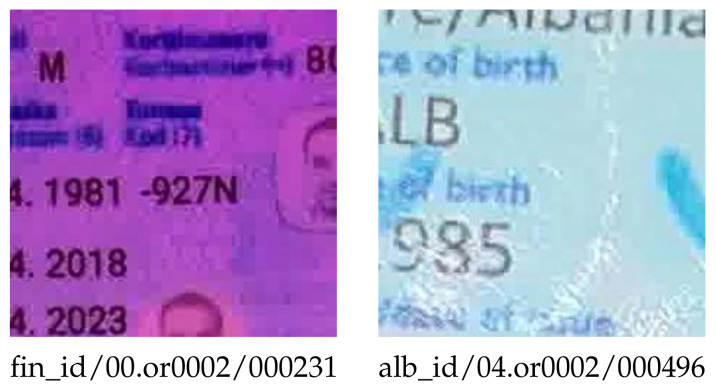
FP error samples for CNN-based screen recapture detector.

**Table 1 jimaging-08-00181-t001:** Document types featured in DLC-2021.

#	Document Type Code	Document Type
1	alb_id	ID Card of Albania
2	aze_passport	Passport of Azerbaijan
3	esp_id	ID Card of Spain
4	est_id	ID Card of Estonia
5	fin_id	ID Card of Finland
6	grc_passport	Passport of Greece
7	lva_passport	Passport of Latvia
8	rus_internalpassport	Internal Passport of Russia
9	srb_passport	Passport of Serbia
10	svk_id	ID Card of Slovakia

**Table 2 jimaging-08-00181-t002:** Types of video.

Video Type Code	Description
cc	unlaminated color copy
cg	unlaminated gray copy
or	“original” laminated documents from MIDV-2020 collection
re	video recapture for document on device screen

**Table 3 jimaging-08-00181-t003:** Smartphone camera specification.

Characteristic	iPhone XR [27]	Samsung S10 [28]
focal length (equivalent)	26 mm	26 mm
lens aperture	f/1.8	*f*/1.5–2.4
sensor size	1/2.55”	1/2.55”
sensor pixel size	1.4 μ	1.4 μ
sensor pixel count	12 MP	12 MP

**Table 4 jimaging-08-00181-t004:** Number of video clips by type.

Smartphone	Resolution	FPS	Video Type				Total
			or	cc	cg	re	
Samsung S10	3840 × 2160	30	140	283	121	200	744
iPhone XR	3840 × 2160	60	70	201	51	200	522
Samsung S10	1920 × 1080	30	40		39		79
iPhone XR	1920 × 1080	30	40		39		79
Total:			290	484	250	400	1424

**Table 5 jimaging-08-00181-t005:** Performance comparison between CNN-based and Scikit Dummy Classifier detectors for a screen recapture per-frame detection task.

Metrics	CNN	Dummy Classifier			
		const = false	const = true	Stratified	Uniform
accuracy	89.67%	59.63%	40.37%	51.28±0.38%	50.16±0.31%
precision	85.89%	–	40.37%	40.29±0.45%	40.53±0.29%
recall	89.03%	0.00%	100.00%	42.93±0.59%	50.20±0.34%

**Table 6 jimaging-08-00181-t006:** CNN-based unlaminated color copy detector architecture (layers).

#	Type	Parameters	Output Size	Activation Function
1	Conv	8 filters 3×3, stride 1×1, no padding	74 × 74 × 8	relu
2	Conv	16 filters 3×3, stride 1×1, no padding	72 × 72 × 16	relu
3	MaxPool	pooling 2×2, no padding	36 × 36 × 16	
4	Conv	16 filters 3×3, strides 2×2, padding 2×2	34 × 34 × 16	relu
5	Conv	24 filters 3×3, stride 1×1, padding 1×1	16 × 16 × 24	relu
6	Conv	32 filters 2×2, stride 1×1, padding 1×1	15 × 15 × 32	relu
7	MaxPool	pooling 2×2, no padding	7 × 7 × 32	
8	Conv	32 filters 2×2, stride 1×1, no padding	6 × 6 × 32	relu
9	Conv	12 filters 2×2, stride 1×1, no padding	5 × 5 × 12	relu
10	Flatten		1 × 1 × 300	
11	Dropout	dropout rate = 0.4	1 × 1 × 300	
12	Fully Connected	2 outputs	1 × 1 × 2	softmax

**Table 7 jimaging-08-00181-t007:** Performance comparison between CNN-based and Scikit Dummy Classifier detectors for an unlaminated color copy per-frame detection task.

Metrics	CNN	Dummy Classifier			
		const = false	const = true	Stratified	Uniform
accuracy	83.61%	8.92%	91.08%	74.39±0.19%	49.98±0.21%
precision	96.01%	–	91.08%	91.10±0.07%	91.07±0.15%
recall	85.56%	0.00%	100.00%	79.67±0.20%	49.99±0.17%

**Table 8 jimaging-08-00181-t008:** Performance of the Scikit Dummy Classifier in a gray copy per-frame detection task.

Metrics	Dummy Classifier			
	const = false	const = true	Stratified	Uniform
accuracy	60.83%	39.17%	44.06±0.25%	50.09±0.26%
precision	–	39.17%	39.22±0.15%	39.26±0.25%
recall	0.00%	100.00%	77.82±0.32%	50.11±0.35%

## Data Availability

The data presented are openly available under Creative Commons Attribution–ShareAlike 2.5 Generic License (CC BY-SA 2.5) in Zenodo at doi:10.5281/zenodo.6466767, doi:10.5281/zenodo.6466770, doi:10.5281/zenodo.6466764.

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
