# Peer review of "Document Liveness Challenge Dataset (DLC-2021)"

_2313-433X, 2022, doi:10.3390/jimaging8070181_

Round 1

Reviewer 1 Report

There are several major concerns:
1) Authors should further clarify and elaborate novelty in their contribution.
2) More discussion on results needed and conclusion is short
3) What are the practical implications of this research.
4) References format is not as per journal requirement.
4) Regarding the machine learning, image classification, more literature needed to give a provide stronger backing to your claims. Check the below work: -
* XGBoost: 2D-Object Recognition Using Shape Descriptors and Extreme Gradient Boosting Classifier, Computational Methods and Data Engineering: Proceedings of ICMDE 2020
* 2D Object recognition techniques: state-of-the-art work, Archives of Computational Methods in Engineering, 1-15, 2020
* An efficient technique for object recognition using Shi-Tomasi corner detection algorithm, Soft Computing, 1-10,2020
* An efficient content based image retrieval system using BayesNet and K-NN, Multimedia Tools and Applications 77 (16), 21557-21570
*https://journals.plos.org/plosone/article?id=10.1371/journal.pone.0238058
*Towards Face Presentation Attack Detection Based on Residual Color Texture (https://doi.org/10.1155/2021/6652727)
*Multi-modal Face Anti-spoofing Attack Detection Challenge at CVPR2019
* Content-based image retrieval system using ORB and SIFT features, Neural Computing and Applications 32 (7), 2725-2733
* Underwater image enhancement using blending of CLAHE and percentile methodologies, Multimedia Tools and Applications 77 (20), 26545-26561
* Underwater image enhancement using blending of CLAHE and percentile methodologies, Multimedia Tools and Applications 77 (20), 26545-265615) equations are not in proper format equation numbers are missing.
6) The inspiration of your work must be highlighted. I would suggest adding some recent literature in the manuscript.

Total paper is not in proper format.

Author Response

We carefully take into consideration all comments.

1) Authors should further clarify and elaborate novelty in their contribution.
We present the first public GPDR compliant dataset with recaptured and grey-copy ID documents.

2) More discussion on results needed and conclusion is short
We corrected a conclusion to reflect the main results of presenting a new data set. What aspects of the presented data set do you consider important to discuss in more detail?

3) What are the practical implications of this research.
Authors believe that the provided dataset will serve as a valuable resource for ID document recognition and ID document fraud prevention research community and lead to more high-quality scientific publications in the field of ID documents analysis, as well as in the general field of computer vision.

4) References format is not as per journal requirement.
MDPI LaTeX template and BibTeX reference data are used for manuscript preparation. Please provide examples of reference numbers that do not meet the requirements of the journal  

4) Regarding the machine learning, image classification, more literature needed to give a provide stronger backing to your claims. Check the below work:
We have added links to relevant recent works on the subject of the article.

5) equations are not in proper format equation numbers are missing.
Please specify which equations do not match the format and specify the page numbers (or sections) on which there are missing equation numbers.

6) The inspiration of your work must be highlighted. I would suggest adding some recent literature in the manuscript.
We have added links to relevant recent works on the subject of the article. 

Reviewer 2 Report

The database presented in this article will certainly be an important tool to advance research on the detection of fraud based on the presentation of copies of documents. The extension of the database with video clips is very important as well as the itroduction of different types of variations in the acquisition. I would suggest that the authors make it more explicit whether the database is GDPR compliant or not. If yes, I would suggest to mention it in the abstract as well.

As the videos were captured with two different smartphones, this allows the application of techniques based on the study of sensor noise (PRNU). This could be mentioned in the future research section.

Author Response

Thank you for your attention and valuable suggestions that helped make the text of the article better.

1. DLC-2021 complies with the GDPR and we have added this meaningful information to the abstract.
2. Sensor noise (PRNU) based methods are mentioned in the future research section.

Reviewer 3 Report

The manuscript presents a challenge dataset for ID document forensics tasks.  There are 1424 video clips captured in the real-word conditions. It is a topic of interest to researchers in the document forensics areas but several issues still need to be addressed. 

A. It is not clear about the camera setting for video capture. The camera setting is an important factor for video forensics, and provides useful information for video analysis. However, the manuscript only give the information about resolution and FPS, miss the other information.

B. the authors captured the video using two smartphones, which places a limitation on the diversity of the dataset. 

C. the baseline methods should be evaluated on this dataset. 

D. The introduction is need to be extended. Some related works are missed. For example, C. Chen, S. Zhang, and F. Lan, “A Database for Digital Image Forensics of Recaptured Document,” arXiv preprint arXiv:2101.01404, 2021.

E. The recaptured documents from screen shot show the obvious moire pattern as shown in Fig. 7, which could decrease in value of the dataset. The recaptured document with a high quality is expect.  In addition, the screen shot from smartphone screen could be more important. 

Author Response

Thank you for your attention and valuable suggestions that helped make the text of the article better.

  1. Table 3 with smartphone camera specification added. On the other hand the main purpose of the DLC-2021 is to simulate checking a user-created video, and shooting options in this case are often not available at all.
  2. Yes, such limitation exists and we have a plan to make the dataset more diverse in future works
  3. Section “4. Experimental baselines” added.
  4. The introduction was extended, next related works are cited:
    22. Chen, C.; Zhang, S.; Lan, F.; Huang, J. Domain generalization for document authentication against practical recapturing attacks.arXiv preprint arXiv:2101.014042021.
    23.Yan, J.; Chen, C. Cross-Domain Recaptured Document Detection with Texture and Reflectance Characteristics.  2021 Asia-PacificSignal and Information Processing Association Annual Summit and Conference (APSIPA ASC). IEEE, 2021, pp. 1708–1715.
  5. Complex document background makes the task of screen recapture detection quite difficult even in the presence of such noticeable artifacts. We understand the importance and value of smartphone screenshots,  it is planned for the next versions of the dataset.

Round 2

Reviewer 3 Report

The authors have addressed most of my concerns. 

Author Response

After round 2 the following changes were made:
1. the novelty of the presented data set is indicated in the abstract
2. the experimental data are updated in Section 4 ("Experimental baselines")
3.the contribution of the added author is listed in the section "Author Contributions"
4. fixed minor typos

No more fixes have been made as there is no more detailed list of fixes and round 2 results from other reviewers have not been received.